# The Structural and Electrochemical Properties of CuCoO_2_ Crystalline Nanopowders and Thin Films: Conductivity Experimental Analysis and Insights from Density Functional Theory Calculations

**DOI:** 10.3390/nano13162312

**Published:** 2023-08-11

**Authors:** Hasnae Chfii, Amal Bouich, Andreu Andrio, Joeluis Cerutti Torres, Bernabé Mari Soucase, Pablo Palacios, Mohammed Abd Lefdil, Vicente Compañ

**Affiliations:** 1Escuela Técnica Superior de Ingeniería del Diseño, Universitat Politècnica de València, 46022 València, Spainbmari@fis.upv.es (B.M.S.); 2Instituto de Energía Solar, ETSI Telecomunicación, Universidad Politécnica de Madrid, Ciudad Universitaria, 28040 Madrid, Spainpablo.palacios@upm.es (P.P.); 3Departamento de Física, Universitat Jaume I, 12080 Castellón de la Plana, Spain; andrio@uji.es; 4Departamento Física Aplicada a las Ingenierías Aeronáutica y Naval, ETSI Aeronáutica y del Espacio, Universidad Politécnica de Madrid, Pz. Cardenal Cisneros, 3, 28040 Madrid, Spain; 5Laboratory MANAPSE, University Mohammed V, Rabat 10100, Morocco; 6Departamento de Termodinámica Aplicada, Universitat Politècnica de Valencia, 46022 Valencia, Spain

**Keywords:** delafossite, powder, films, spray pyrolysis, EIS, conductivity, relaxation time

## Abstract

A novel manufacturing process is presented for producing nanopowders and thin films of CuCoO_2_ (CCO) material. This process utilizes three cost-effective synthesis methods: hydrothermal, sol-gel, and solid-state reactions. The resulting delafossite CuCoO_2_ samples were deposited onto transparent substrates through spray pyrolysis, forming innovative thin films with a nanocrystal powder structure. Prior to the transformation into thin films, CuCoO_2_ powder was first produced using a low-cost approach. The precursors for both powders and thin films were deposited onto glass surfaces using a spray pyrolysis process, and their characteristics were examined through X-ray diffraction, scanning electron microscopy, HR-TEM, UV-visible spectrophotometry, and electrochemical impedance spectroscopy (EIS) analyses were conducted to determine the conductivity in the transversal direction of this groundbreaking material for solar cell applications. On the other hand, the sheet resistance of the samples was investigated using the four-probe method to obtain the sheet resistivity and then calculate the in-plane conductivity of the samples. We also investigated the aging characteristics of different precursors with varying durations. The functional properties of CuCoO_2_ samples were explored by studying chelating agent and precursor solution aging periods using Density Functional Theory calculations (DFT). A complementary Density Functional Theory study was also performed in order to evaluate the electronic structure of this compound. Resuming, this study thoroughly discusses the synthesis of delafossite powders and their conversion into thin films, which hold potential as hole transport layers in transparent optoelectronic devices.

## 1. Introduction 

In recent decades, there has been a significant demand for advancements in the integration of high-efficiency p-type transparent conductive oxides (TCOs) into industrial applications, particularly in the past few decades [1,2,3,4,5]. However, the research community is keen on exploring alternative TCOs and enhancing their electrical and optical properties to improve device efficiency [6,7,8]. Among various delafossite oxides, copper-based delafossite oxides stand out as promising candidates due to their desirable electrical and optical characteristics. The copper-based delafossite oxide, specifically CuMO_2_ (where M represents Al, Ga, Fe, Co, Mg, Fe, Cr…), has attracted considerable attention for diverse applications such as batteries [9,10,11,12,13,14,15], luminescent materials [16,17,18,19,20,21], thermoelectrics [22,23,24,25,26,27,28], solar energy conversion and photocatalysis [29,30,31,32,33,34,35], hydrogen production through water splitting using photocathodes [36,37,38,39,40,41], and gas sensors [42,43,44,45,46,47,48,49]. In this study, we focus on the CuCoO_2_ compound, in view of intriguing structural, optical, and electrical properties; it has only been studied in a limited number of publications, and in most of them, the CuCoO_2_ compound has shown interesting properties to be treated as a possible material in photovoltaic solar cells [50,51,52,53,54,55,56,57,58]. Extensive studies on delafossite powders and thin films reveal that CuCoO_2_ is a p-type semiconductor that crystallizes in two distinct structures: rhombohedral (3R) [59,60,61] and hexagonal (2H) [60,62,63]. The successful synthesis of the CuCoO_2_ phase has been reported in only a few papers to date. Moreover, various efforts have been made to manipulate the properties of CuCoO_2_, such as characterizing it using different printing techniques or exploring its physical and chemical properties through doping processes involving a wide range of elements [64,65,66,67]. Considerable attention has been devoted to the integration of this delafossite material into various domains using different methodologies. Beekman et al. published a study on the synthesis of undoped delafossite through ion exchange [59], while Z. Du et al. synthesized it as an electrocatalyst for the oxygen reaction [54,57] and also investigated the solvothermal synthesis of CuCoO_2_ [57,65]. Isacfranklin et al. focused on CuCoO_2_ electrodes for supercapacitor applications [65], whereas D. Xiong et al. conducted a study on the polyvinylpyrrolidone-assisted hydrothermal synthesis of CuCoO_2_ [67]. Other researchers explored hydrogen-related aspects, such as J. Ding et al., who investigated Co_3_O_4_-CuCoO_2_ composites [68].

Several studies have reported on the effects of Ca^+2^ doping in CuCoO_2_. Z. Du et al. studied the optical and electrical properties of the material [57,64,67], while M. Yang et al. examined the impact of nickel doping on the structure and morphology of delafossite [58]. Limited research has been conducted on the transformation of delafossite powder into thin films [52,56,58]. Specifically, there is currently a lack of research projects focusing on the structural and electrochemical properties of CuCoO_2_ crystalline nanopowders and thin films when they are deposited on glass substrates.

Several physical and chemical processes, such as spray pyrolysis [53,69,70,71,72] and spin-coating [73,74,75,76], have been employed to fabricate thin films of CuCoO_2_. However, previous attempts to produce CuCoO_2_ thin films have predominantly utilized complex and expensive techniques, posing significant practical limitations. Currently, only a few endeavors have focused on generating CuCoO_2_ thin films in the hexagonal phase, particularly through chemical deposition methods. Therefore, the objective of this study is to develop a simpler and more efficient chemical synthesis using three different methods based on the utilization of CuCoO_2_ powder, which yields an excellent, ordered crystalline mixture between two structures: rhombohedral and hexagonal with and good band gap values aiming to achieve thin films of this material. Comprehensive investigations were conducted to thoroughly analyze the optical and electrical characteristics of these thin films. In particular, studies of electrochemical impedance spectroscopy (EIS) analysis and simulation of delafossites-based solar cells where CuCoO_2_ acting as low temperature hydrothermal (HTL) material is reported for the first time. The conductivity of samples, prepared using the three different techniques, was measured in direct current (named in-plane conductivity) using the four-point probe and alternating current (named transversal conductivity) by electrochemical impedance spectroscopy. 

Additionally, Density Functional Theory was employed to provide a theoretical understanding of the role played by each element in the electronic structure, as well as to accurately determine the bandgap of the material.

## 2. Experimental Part

### 2.1. Synthesis of CuCoO_2_ Powders

CuCoO_2_ was produced using three different methods. The grains of copper(II) nitrate trihydrate (Cu(NO_3_)_2_, 3H_2_O; 99%) were used as the Cu^+2^ source; cobalt(II) nitrate hexahydrate (Co(NO_3_)_2_, 6H_2_O; 98%) was used as the Co^2+^ source; and sodium hydroxide (NaOH) was obtained from Sigma–Aldrich. As a solvent, deionized water (DW) was utilized. At room temperature, all chemicals were added. The amount of each precursor was 2 mM of each of the copper and cobalt sources, 4.40 g of sodium hydroxide, and 70 mL of deionized water (DW), respectively. All of the precursors were measured and mixed in a reasonable amount of solvent for three hours. For hydrothermal synthesis: method 1 (sample named CuCoO_2__H), the liquid was placed in a 100 mL Teflon autoclave and autoclaved at 100 °C for 24 h; we washed the obtained solution several times with distilled water. Method 2: solid-state reaction (sample named CuCoO_2__SSR); stoichiometric amounts of the above-mentioned powders were ground with ethanol solution (95%) for 24 h. The ground powder was calcined at 800 °C for 5 h. Method 3: sol-gel (sample named CuCoO_2__SG); copper(II) nitrate and cobalt(II) nitrate were mixed in ethylene glycol. The solution was agitated at room temperature for 1 h in a beaker before being dried at 150 °C for 5 h. On a heated plate with a magnetic stirrer, gelation took place until a purple color emerged. The amorphous powder was heated incrementally from 50 °C to 800 °C.

### 2.2. Synthesis of the CuCoO_2_ Thin Films

The glass substrate cleaning process was performed by ultrasonically cleaning and drying the substrate. The dissolved nanocrystal precursor solution was then converted into a thin film. CuCoO_2_/glass films were produced using spray pyrolysis technology. First, we dispersed 1 mg of CuCoO_2_ powder in 10 mL of ethanol/water mixture and sonicated for 30 min to form homogeneous slurry. The CuCoO_2_ slurry was then deposited on the glass film (Figure 1). The resulting film was annealed in air at 350 °C for 40 min to form a CuCoO_2_-coupled glass film.

### 2.3. The Four-Probe Method

The measurement of conductivity in powders and polymeric thin membranes is a complex task that is influenced by various factors, including sample casting preparations, thermal/hydrothermal treatments, relative humidity, and the cell configuration used for film resistance measurements, as well as the pressure applied between the probe electrodes [77]. The four-probe method is commonly used to measure sample sheet resistance and estimate in-plane conductivity, but it may result in inaccuracies, particularly for materials with morphological anisotropy [78,79]. According to this method in which four probes are arranged equidistantly in a straight line and pushed against the film as shown in Figure 2, the resistivity may be calculated through determining the potential difference between the RE and S electrodes, due to current passing via an easily identifiable connection between the WE and CE electrodes (Figure 2). Knowing the resistivity, the sheet conductivity can be calculated from the inverse of the sheet resistivity as
(1)$$\sigma =\frac{1}{\rho }=\frac{1}{R_{s}\cdot t}$$
where *ρ* is the sheet resistivity, *R_s_* the in-plane resistance, and *t* is the sheet sample thickness. In our samples, the thickness values are 140 nm, 136 nm, and 142 nm for CuCoO_2__H, CuCoO_2__SG, and CuCoO_2__SSR, respectively.

### 2.4. Electrochemical Impedance Spectroscopy (EIS) Measurements

The conductivity was measured through the samples with a Novocontrol broadband dielectric spectrometer (BDS) equipped with an SR 830 lock-in amplifier and an Alpha dielectric interface in the frequency interval from 10^−1^ to 10^7^ Hz with 0.1 V amplitude of the signal at temperatures ranging from 20 to 120 °C in increments of 20 °C. The samples were airbrushed before being tested, and their thicknesses were determined via a micrometer, averaging 10 readings from various sections of the surface. The samples were dried in a vacuum cell and placed between two gold round electrodes that served as blocking electrodes before being heated in the Novocontrol system in a neutral nitrogen-free environment. A temperature cycle from 20 to 120 °C in 20 °C increments was performed before collecting the dielectric spectra at every step to ensure uniformity and reduce interference from remaining water. During the testing, the electrodes were kept completely wet below 100 °C and replicated a 100% relative humidity environment above 100 °C in a BDS 1308 liquid device that was attached to the spectrometer and contained deionized water. To accurately control the temperature conditions, the temperature was kept constant throughout the conductivity measures (isothermal investigations) or shifted stepwise from 20 to 120 °C using a nitrogen jet (QUATRO from Novocontrol), alongside a temperature error of 0.1 K throughout each frequency sweep. 

The frequency dependence of complex impedance *Z**(*ω*) = *Z*′(*ω*) *+ j·Z*″(*ω*) yields the real component of conductivity as
(2)$$\sigma^{\prime }\left(\omega \right)=\frac{Z^{\prime }\left(\omega \right)\cdot L}{\left[{\left(Z^{\prime }\left(\omega \right)\right)}^{2}+{\left(Z^{\prime \prime }\left(\omega \right)\right)}^{2}\right]\cdot S}=\frac{L}{R_{0}\cdot S}$$
where *L* and *S* represent the thickness and area of the sample in contact with the electrodes, and *R*_0_ represents its resistance.

## 3. Results and Discussion

### 3.1. Structural Analysis

Figure 3 presents the XRD diffractograms of the as-prepared samples (a) CuCoO_2__H, (b) CuCoO_2__SG, and (c) CuCoO_2__SSR with a mixture of two structures: rhombohedral and hexagonal. This was confirmed by checking the databases: JCPDS Map No. 074-1855 and JCPDS Map No. 021-0256. The three patterns observed in Figure 3 were identified as pure phases of CuCoO_2_ without secondary phases. The main peak of delafossite CuCoO_2_ in the rhombohedral structure (110) is at 2θ = 37.92°, and the hexagonal structure is at 2θ = 38.27°.

After characterization of CuCoO_2_ nanoparticles prepared by three methods: the hydrothermal method, sol-gel method, and solid-state reaction method, we examined their structure following deposition onto glass substrates and thin film synthesis using spray pyrolysis. Figure 4 shows the XRD graphs of the thin films of CuCoO_2_ on the glass substrates. We observed some of the peaks shown in the XRD pattern, which now appear in the diffractograms of the deposited film. However, the structure did not change, and the material showed two structures containing delafossite: 3R-CuCoO_2_ (JCPDS#21-0256) and 2H-CuCoO_2_ (JCPDS#74-1855). Furthermore, it is clear that each XRD pattern begins with a tablet. It belongs to the glass substrate.

### 3.2. FE-SEM Analysis

We are interested in the morphology and particle size distribution of our compound CuCoO_2_. Observations by scanning electron microscopy (SEM) were carried out on a submicron scale on the three prepared CuCoO_2_ nanocrystal powders as shown in Figure 5. A 1 μm magnification is given. 

FE-SEM images of the powders show that the submicron CuCoO_2__H, CuCoO_2__SG, and CuCoO_2__SSR are in the form of a powder made up of crystals of different sizes, and they contain agglomerates of hexagonal particles and crystals like rhombohedral shapes. There have been no additional morphologies detected, which confirms the XRD results.

Figure 6 shows FE-SEM images of produced thin films comprising CuCoO_2__H, CuCoO_2__SG, and CuCoO_2__SSR at 1 μm. After deposition with the spray pyrolysis technique, all films had a uniform distribution of nanocrystalline particles.

### 3.3. HR-TEM

#### Analysis

Figure 7 shows the characterization of the three samples with transmission electron microscope (TEM) characterization, and the high-resolution transmission electron microscopy (HR-TEM) images of CuCoO_2_ are illustrated in Figure 7. As a result, we could assume that the crystallinity of CuCoO_2_ retains a structure mostly constituted of nanocrystals smaller than 15 nm in diameter. HR-TEM d-spacings are likewise consistent with a mixture of rhombohedral and hexagonal CuCoO_2_ chalcopyrite phases. The results from the FE-SEM agree with the HR-TEM images, which show clearly established small grains of some tens of nanometers.

Figure 8 depicts the transmission electron microscopy (HR-TEM) mapping. The photos of the three CuCoO_2_ powders reveal a good distribution of copper–cobalt oxide components, with the hydrothermal CuCoO_2_ powder having a higher crystallinity.

## 4. EDX Analysis

We further investigate the chemical composition of our sample using the EDS technique. In Figure 9 and tables below we show the results of the EDS analysis for the cracked surfaces of the CuCoO_2__H, CuCoO_2__SG, and CuCoO_2__SSR samples. From the table above, the percentages of each Cu and Co in a position are approximately the same, but the percent of oxygen is a bit high. This difference may be due to the oxidation of copper or cobalt.

## 5. Optical Properties

The transmittance spectra have been used to examine optical qualities. In the visible area, all of the CuCoO_2_ thin films displayed extensive absorption. The absorbance coefficients for thin film samples generated utilizing a spray pyrolysis method containing nanocrystals were determined. In the transmittance spectrum of materials, the absorption coefficient is connected with the optical energy gap or it is in the strong absorption zone, which may be estimated using Tauc’s equation [74]
(3)$$\alpha =\frac{A{\left(h\nu -E_{g}\right)}^{n}}{h\nu }$$

*A* is a constant, *h* is the Planck constant, *ν* is the frequency, and n is an indicator of the optical absorption process. It equals 2 for directly permitted transitions and 0.5 for indirectly permitted transitions. Figure 10 shows (a) the transmission values and (b) the Tauc plot for CuCoO_2_ thin films. The arrangement of Tauc’s figure suggests that the CuCoO_2_ thin film under deposit has a straight band gap. *E_g_* may be estimated by extrapolating by projecting a horizontal line to the point of zero absorption coefficient (*α* = 0). The band gaps were calculated by graphing (*αhν*)^2^ vs. energy in eV and extrapolating the linear portion of the spectrum (*hν*). According to the transmission graph (Figure 10a), the powder CuCoO_2__SG has the highest transmission value compared to the two other powders CuCoO_2__H and CuCoO_2__SSR. The order of the delafossite powders transmission is obviously confirmed with the band gap graphs (Figure 10b). 

In the Tauc’s plot, the “linear part” is selected by examining the absorption data at higher photon energies, where the absorption is predominantly governed by indirect transitions, with its absorption coefficient being practically constant. In our study, the calculation of band gap values has an error of ±0.10 eV and was obtained by taking the linear part of the curve (between 4 and 4.5 eV), fitting these points to a straight line, and extrapolating this line until it intersects the base line (OX axis). The intersection value (in eV) is the direct band gap according to Tauc’s model [74]. In our study, the values obtained for our delafossite material are 3.51 ± 0.10 eV, 3.77 ± 0.12 eV, and 3.87 ± 0.10 eV for CuCoO_2_-H, CuCoO_2_-SSR and CuCoO_2_-SG, respectively. Depending on the number of points chosen in the range considered, it works as if it were a transparent layer with a certain uncertainty. This suggests that delafossite is likely to be a good transmitter of charge carriers, leading to a higher band gap value of around 3.5 eV.

## 6. In-Plane Conductivity Measurements

In powders used for solar cells, the four-point probe method is the most often used method for assessing the electrical characteristics of conducting films [78,79]. This approach has been utilized to evaluate the in-plane conductivity of CuCoO_2_ films on top of non-conductive substrates (in our instance, glass), which are typically created by spray pyrolysis technology of the dispersions employed in this work. The experimental procedure used is the following. The four-point probe is attached to a source meter that supplies a certain current. A source meter’s current (I) flows through the two outer probes, and a voltammeter can measure the voltage (V) across the two inner probes. By plotting the voltage measured for each current intensity, the sheet resistance, Rs, can be determined, as is shown in Figure 11. 

A close inspection of these figures reveals that sample resistance (*R_s_*) of the CuCoO_2_ films is constant, and its values can be obtained from the slope of the experimental fit determined from the plot of the voltage versus intensity, where a clear linearity is observed for all the samples. According to the KIT used to measure the resistance of the film by means of the four-points method, the value of *R_s_* is given by
(4)$$R_{s}=4.532\times \frac{V}{I}$$

The value for said KIT as a consequence of the geometry used in the measurement is the constant 4.532. Therefore, the in-plane conductivity, given in Equation (3), is determined from the sheet resistance from Equation (4). The results of the three produced CuCoO_2_ samples using the four-point probe technique and measured at ambient temperature are presented in Table 1. The conductivity values change with the chelating agent and the aging duration. These findings imply that aging period and thickness change have an effect on electrical conductivities.

## 7. Dielectric Spectra Analysis

The electrical impedance spectroscopy (EIS) measurements were performed on all samples to determine the conductivity measured in the transversal direction, named direct current conductivity (σ_dc_). Such measurements were carried out over a temperature interval of 20 °C to 120 °C in two steps to ensure reproducibility within the temperature interval. The experimental data obtained for the samples from the Novocontrol were examined to obtain the complex dielectric permittivity function, denoted as ε*(ω,T); and the complex conductivity function, denoted as σ*(ω,T), where j is the imaginary unit (j^2^ = −1), ε_0_ is the vacuum permittivity, and ω is the angular frequency of the electric field that was applied (ω = 2πf). Different methods have been used to determine the dc-conductivity of a sample from dielectric spectroscopy data analysis [80,81,82,83,84,85,86,87,88,89,90]. In this work, we have used the Bode diagram obtained from the complex dielectric spectra, where the complex conductivity is given by σ*(ω,T) = j ε_0_ ω ε*(ω,T), which can be expressed in terms of the real and imaginary part, σ′(ω,T) and σ″(ω,T), respectively, and the direct current conductivity σ_dc_ was calculated [91,92,93,94]. This technique was used in this study to examine data for the real component of conductivity in dry conditions by graphing conductivity (in S cm^−1^) vs. frequency (in Hz) using the appropriate Bode diagrams for all temperature ranges.

The Bode diagrams for CuCoO_2__H, CuCoO_2__SG, and CuCoO_2__SSR delafossite materials were investigated at temperatures ranging from −20 °C to 120 °C, with increments of 20 °C, as shown in Figure 12. Additional graphs demonstrating the variation of phase angle (ϕ) vs. frequency at identical temperatures are included in the Appendix A. Upon closer examination of the figures, it can be observed that the conductivity tends to a constant value (plateau) when the phase angle (ϕ) approaches zero or reaches a maximum, indicating the direct-current conductivity (σ_dc_) of the sample. Furthermore, a decrease in conductivity with decreasing frequency was observed in the high-frequency region, along with a transition zone where the cut-off frequency ranges from 10^5^ to 10^6^ Hz for CuCoO_2__SG and CuCoO_2__SSR samples and starts increasing with frequency. In the case of the hydrothermal sample CuCoO_2__H, the real part of conductivity remains constant at low frequencies until a cut-off frequency between 10^3^ Hz and 10^6^ Hz, after which it starts increasing with frequency. The initial process is connected to the sample’s resistance/stability, but this second process is connected to the dispersion (charge transfer) caused by the charge’s mobility, as the sample behaves like a capacitor. The conductivity values presented were derived using the peak frequency when the phase angle approaches 0.

Upon careful examination of Figure 12, it can be observed that samples CuCoO_2__SG and CuCoO_2__SSR exhibit a nearly constant conductivity across a wide range of frequencies and temperatures; that is standard behavior for a conductive material. Identical behavior has been reported in previous studies on nanocomposites of multilayer graphene in polypropylene [95]. This phenomenon is due to Debye relaxation, which occurs as a result of the mobility and redirection of dipoles and localized charges at high frequencies in response to an applied electric field and dominates direct-current conductivity [91,92,96]. The change in dc-conductivity of the samples at various temperatures may be determined from the plateau where the phase angle is zero or tends to zero. For frequencies where the phase is near zero, we have a pure resistive impedance that can be attributed to the ionic conductivity alone. This value is the active phase with high-efficiency electrochemical processes and respects the charge transport into the powders. These phenomena are observed for all the samples at frequencies below 10^4^ Hz. Moreover, with rising temperature, the frequency at which the point of equilibrium occurs moves to high frequencies, where a plateau in the Bode diagram from low to high frequencies can be observed, suggesting thermal activation of ionic transport. The constant value of conductivity suggests that the sample solely shows resistive contribution, and the quantity measured represents the sample’s electrical conductivity. For example, Figure 12 shows that at 20 °C, the trough plane conductivity values followed the trend: σ_CuCoO2_SG_ (5.2 × 10^−5^ S cm^−1^) > σ_CuCoO2_SSR_ (2.9 × 10^−5^ S cm^−1^) > σ_CuCoO2_H_ (4.53 × 10^−8^ S cm^−1^). Similar trends can be observed for the other temperatures studied (for example, 40 °C, 60 °C, 80 °C, 100 °C and 120 °C). For 120 °C the conductivity values obtained, follow the trend, σ_CuCoO2_SG_ (1.4 × 10^−3^ S cm^−1^) > σ_CuCoO2_SSR_ (5.7 × 10^−4^ S cm^−1^) > σ_CuCoO2_H_ (1.0 × 10^−5^ S cm^−1^), respectively. Among the three samples, the greatest proton conductivity of about 10^−3^ S cm^−1^ at 120 °C was found for the CuCoO_2__SG sample and was one order of magnitude higher than CuCoO_2__SSR, where excellent ionic conductivities of about 10^−4^ S cm^−1^ were also shown. These results showed that the preparation method used is very relevant to obtaining excellent results in the measured transversal conductivity; in our case, around of one order of magnitude better than sample CuCoO_2__H was reached. All these values have better conductivities than CIGS:Cr crystalline nanopowders and CuInGaSe_2_ (CIGS) chalcopyrite thin films doped with Cr in varying concentrations [93]. 

From the plot shown in Figure 13, we observe that dc-conductivity increases with the increase of temperature of all mixtures, following an Arrhenius behavior for the thermal activation energy. The measurements of the activation energy calculated from the slopes follow the trend E_act_ (CuCoO_2__SSR) = 27.4 kJ/mol < E_act_ (CuCoO_2__SG) = 30.8 kJ/mol < E_act_ (CuCoO_2__H) = 52.3 kJ/mol, respectively. 

These results indicate that the thin films prepared from hydrothermal synthesis have higher activation energy and lower conductivities than the samples prepared from (a) a solid-state reaction where stoichiometric amounts of the above-mentioned powders have been ground with ethanol solution and, after, calcined at a temperature of 800 °C for 5 h. (b) sol-gel; copper(II) nitrate and cobalt(II) nitrate were mixed in ethylene glycol, and after the gelation occurred, it was placed on a magnetic stirring hot plate until a purple color appeared and the substance become darker. The amorphous powder was heated until 800 °C by steps of 50 °C.

Figure 14 shows the relationship between the relaxation time obtained from the cut-off frequency where the conductivity changes from a constant value in the Bode diagram (plateau) to increasing with frequency increase for all temperatures. In such circumstances, such as is observed here in the case of CuCoO_2__H (see Figure 12c) for all temperatures, according to the power low model, the real part of the conductivity σ′(ω,T) can be expressed in terms of dc-conductivity σ_dc_ and the hopping diffusion rate of protons ω_H_ ≈ 1/τ (in this case) as [94]
(5)$$\sigma^{\prime }\left(\omega ,T\right)=\sigma_{\mathrm{dc}}\left[1+{\left(\frac{\omega }{\omega_{H}}\right)}^{n}\right]$$
where *n* is an exponent with a value between 0 and 1 and is related to interactions between mobile ions (H^+^ in our case) and the dimensionally of the conduction pathway [97]; for instance, this occurs in polymer electrolytes of P[VBTC][Cl]_80_-ran-PMMA_20_ at different temperatures (303 K to 363 K) and P[VBTC][TFSIl]_80_-ran-PMMA_20_ at (308 K to 378 K), respectively [98]. From the fit of the real part of the conductivity shown in the Bode diagrams in Figure 12, we have obtained the values of sample relaxation time; these values are plotted in Figure 14 for each temperature.

From Figure 14, we can see that the relaxation time follows an Arrhenius behavior in all the powders studied, but it is interesting to observe that samples prepared using method 2 (CuCoO_2__SSR) and method 3 (CuCoO_2__SG) have a relaxation time around one order of magnitude smaller than the sample prepared using he method 1 (CuCoO_2__H). This means that the method to produce powders to build thin films is very important to determine their optical and electrical properties. In these results, we observe that delafossites are extremely sensitive to a wide variety of parameters: in particular, the method used in its preparation. The results indicate that these materials have potential in thin film solar cells.

## 8. Theoretical Insight

Theoretical simulations are critical in the understanding of the properties of systems at the atomic level. Therefore, first-principles calculations were performed to complement the experimental results with an insight on the electronic structure and properties. For this purpose, the atomic positions were optimized, and the electronic properties were computed within the Density Functional Theory (DFT) approach. This was conducted based on the framework of the generalized Kohn–Sham scheme [99,100,101] in combination with the projector augmented-wave (PAW) method [102] and the Heyd–Scuseria–Ernzerhof hybrid functional with the modified fraction of screened short-range Hartree–Fock exchange (HSE06) [103,104,105] as implemented in the Vienna ab initio simulation package (VASP) [106,107,108,109,110]. Hybrid functionals allow for a more accurate description of electronic properties of some systems compared to simple DFT calculations with a generalized gradient approximation for the exchange and correlation term in the Kohn–Sham scheme. The higher accuracy, however, is reached at the cost of a higher computational time needed to achieve convergence. 

To model the ${\mathrm{CuCoO}}_{2}$ in the tetragonal phase, a unit cell with 12 atoms (Co_3_Cu_3_O_6_) was used, while for the hexagonal cell, a smaller cell with 8 atoms (Co_2_Cu_2_O_4_) was needed. The electronic wave functions were expanded in a plane wave basis setup to a kinetic energy cutoff of 400 eV. The atomic positions were optimized using the conjugate gradient method up until the forces on each atom were less than 0.01 eV A^−1^; and the energy convergence was less than 10^−8^ eV for the optimization and less than 10^−5^ eV for calculations with the hybrid functional. For the Brillouin zone integration, a 12 × 12 × 6 Monkhorst–Pack scheme k-point mesh was used [111,112,113,114] both for the optimization and the electronic structure calculation.

Figure 15 shows the atom positions and geometrical structure of the converged unit cells. The delafossite structure of this ternary oxide can be appreciated fairly. In the tetragonal structure, an alternate stacking of O-Cu-O dumbbells lie parallel to the *z* axis, and there is a layer of Co-centered octahedrons in the *xy* plane. The stacking follows an ABCABC pattern, forming a trigonal system with lattice parameters a = 2.86 Å and c = 16.98 Å, corresponding to a rhombohedral Bravais lattice of volume 120.1 Å^3^, which matches the experimental data [52,99]. In the hexagonal cells, the same layers of O-Cu-O dumbbells and Co-centered octahedron are observed, but with an ABAB stacking pattern. The lattice constant a = 2.83 Å is practically identical, but c = 11.30 Å is significantly lower; the unit cell volume of 78.47 Å^3^ is also lower. However, these differences are due to the fact that a smaller distance is enough to represent the structure because of the stacking sequence. Hence, important distances within the unit cell, such as the distances between O atoms in the O-Cu-O dumbbells (3.71 Å for the tetragonal 3.69 Å in the hexagonal) or the distances from a cornered O to a centered Co in the octahedron (1.92 Å for the tetragonal and 1.91 Å in the hexagonal) are practically the same.

The band structure and density of states for the tetragonal phase can be appreciated in Figure 16. Though the inclusion of the hybrid functional allows the possibility of avoiding the typical underestimation of gaps in pure DFT approaches; the bandgap values obtained seems to remain underestimated: 2.31 eV for the tetragonal structure and 2.34 for the hexagonal one (experimental results are between 2.5 to 3.65 eV [77,98,103]). However, the structure of the bands and the form of the density of states of the curves are usually well described in this type of calculations, and despite the small difference in bandgap values for both structures, the form of the density of states is practically the same, so only one of them is plotted. 

The obtained gap is an indirect one, which can be appreciated by simple inspection of the bands structure in the left part of Figure 16. The right part shows the density of states in which it is appreciable that the edge of the covalent band is apparently dominated by Cu-3d states. O states show a wide dispersion through the band, but the presence of Co-3d states is high in this region and, except for the peak at the edge of the band, is comparable with the contribution from Cu states. Also, the first peak in the conduction band is mainly due to Co-3d states followed by a second peak with a structure similar to the one that can be found in ${\mathrm{CuAlO}}_{2}$. and ${\mathrm{CuGaO}}_{2}$, as it has been previously reported [98,99,103].

## 9. Conclusions

In summary, the preparation of delafossite CuCoO_2_ using three cost-effective techniques yields a well-ordered crystalline mixture comprising two structures: rhombohedral and hexagonal, exhibiting similar properties and favorable band gap values. To ensure the development of an effective coating solution, careful treatment of the resultant nano-sized precursor powder is necessary. The desired thin films, namely CuCoO_2__H (produced via hydrothermal method), CuCoO_2__SG (produced via sol-gel method), and CuCoO_2__SSR (produced via solid-state reaction method), were successfully obtained through the spray pyrolysis process. Electrochemical impedance spectroscopy measurements reveal that the Sol-Gel method yields films with superior conductivity compared to the other preparation methods. As a result, CuCoO_2_ thin films hold significant potential for solar cell applications. This is supported by the examination of the electronic properties of the rhombohedral CuCoO_2_ through theoretical simulations. Furthermore, electrochemical impedance spectroscopy measurements demonstrate that the copper cobalt delafossites prepared using different synthesis methods have the potential to serve as semiconductor materials. The conductivity values, measured through the plane, increase with temperature as expected: σ_cucoo2_sg_ > σ_cucoo2_ssr_ > σ_cucoo2_h_. Among the three samples, CuCoO_2__SG exhibits the highest conductivity of approximately 10^−3^ S cm^−1^ at 120 °C, which is one order of magnitude greater than CuCoO_2__SSR, while still maintaining good ionic conductivities of approximately 10^−4^ S cm^−1^. This value is approximately two orders of magnitude higher than that of CuCoO_2__H. These conductivity measurements, obtained through impedance spectroscopy, are in agreement with the values determined using the four-probe method, where resistivity follows the trend of ρ_CuCoO2_SG_ < ρ_CuCoO2_SSR_ ≈ ρ_CuCoO2_H_. Lastly, our study demonstrates that samples prepared using method 2 (CuCoO_2__SSR) and method 3 (CuCoO_2__SG) have relaxation times approximately one order of magnitude smaller than samples prepared using method 1 (CuCoO_2__H). This result emphasizes the significant impact that the synthesis method and sample preparation can have on producing powders for building thin films to enhance optical and electrical properties, particularly for solar cell applications.

## Figures and Tables

**Figure 1 nanomaterials-13-02312-f001:**
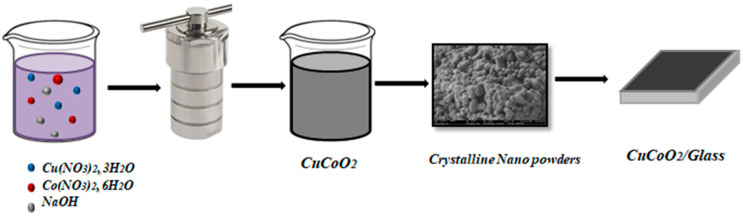
Schematic description of the synthesis process of the CuCoO_2_/Glass film.

**Figure 2 nanomaterials-13-02312-f002:**
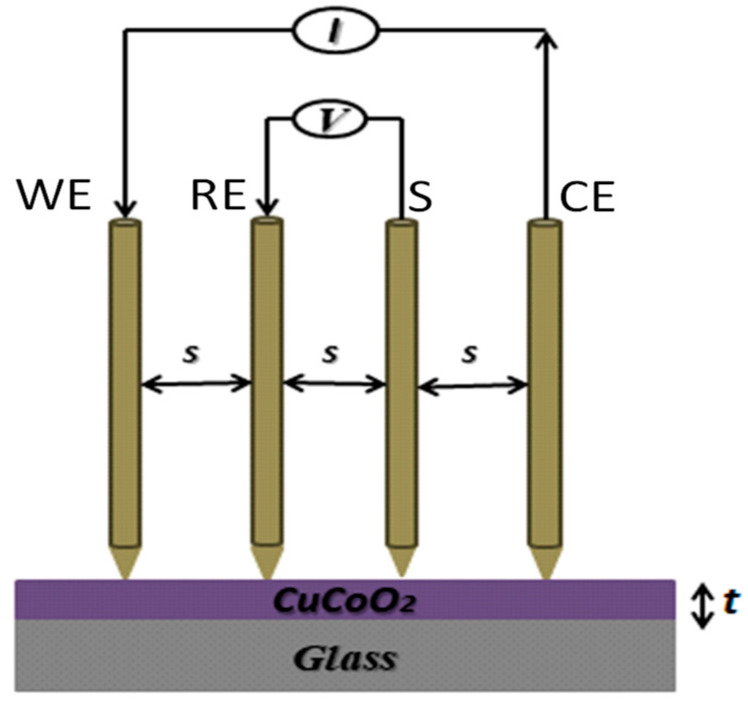
Schematic diagram of the four-point probe method.

**Figure 3 nanomaterials-13-02312-f003:**
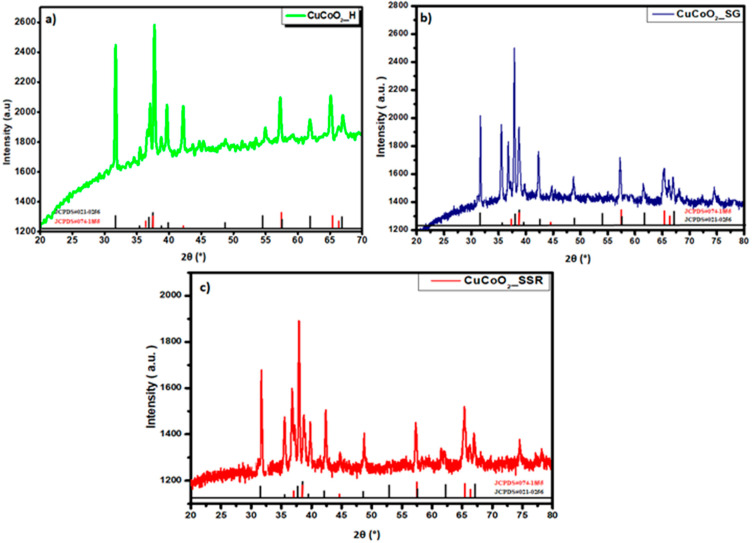
XRD patterns of CuCoO_2_ nanocrystals prepared by (**a**) hydrothermal (CuCoO_2__H), (**b**) sol-gel (CuCoO_2__SG), and (**c**) solid-state reaction (CuCoO_2__SSR).

**Figure 4 nanomaterials-13-02312-f004:**
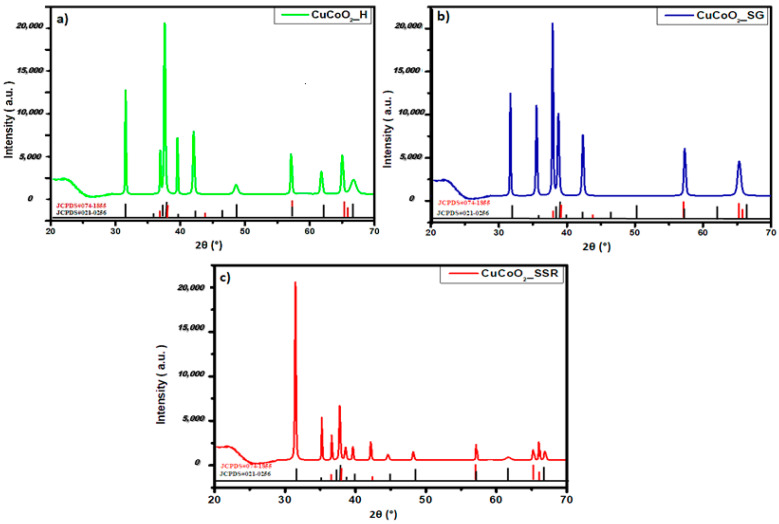
XRD diffractograms of films deposited on glass substrates of (**a**) CuCoO_2__H, (**b**) CuCoO_2__SG, and (**c**) CuCoO_2__SSR.

**Figure 5 nanomaterials-13-02312-f005:**
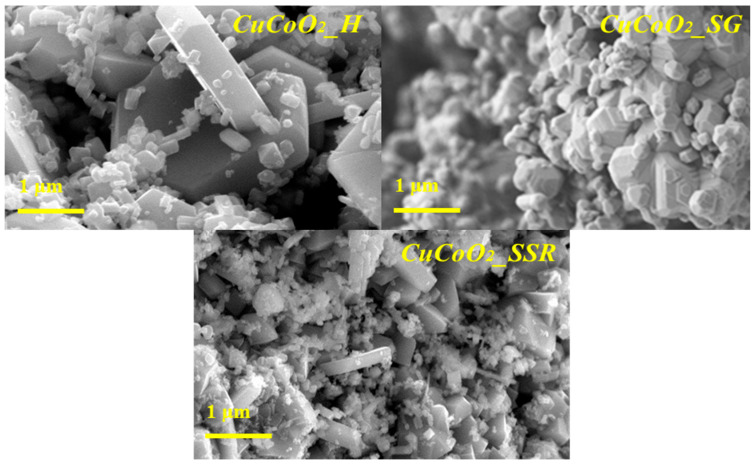
FE-SEM images for the three nanocrystalline powders.

**Figure 6 nanomaterials-13-02312-f006:**
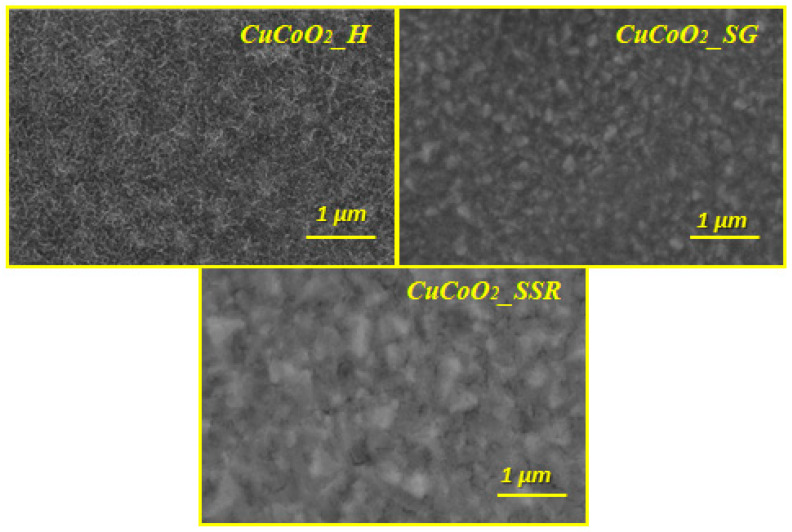
FE-SEM images for CuCoO_2__H, CuCoO_2__SG, and CuCoO_2__SSR films at 1 μm.

**Figure 7 nanomaterials-13-02312-f007:**
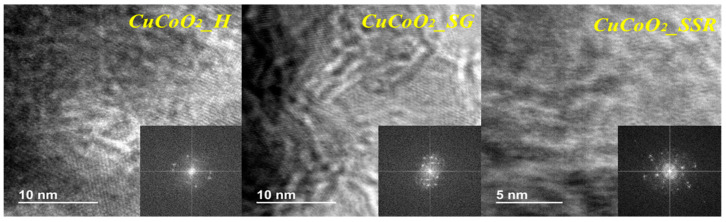
HR-TEM images of CuCoO_2__H, CuCoO_2__SG, and CuCoO_2__SSR.

**Figure 8 nanomaterials-13-02312-f008:**
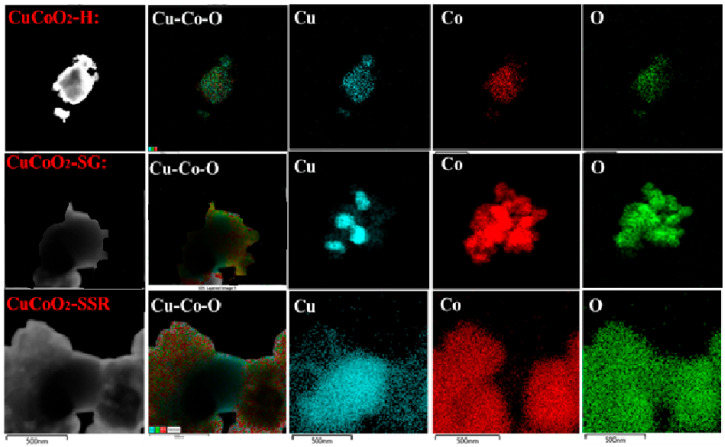
Element mapping analysis of CuCoO_2_ with TEM.

**Figure 9 nanomaterials-13-02312-f009:**
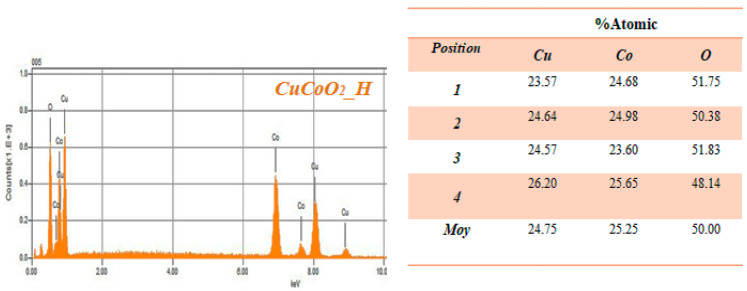
Cracked surface test results of three CuCoO_2_ powders using EDS.

**Figure 10 nanomaterials-13-02312-f010:**
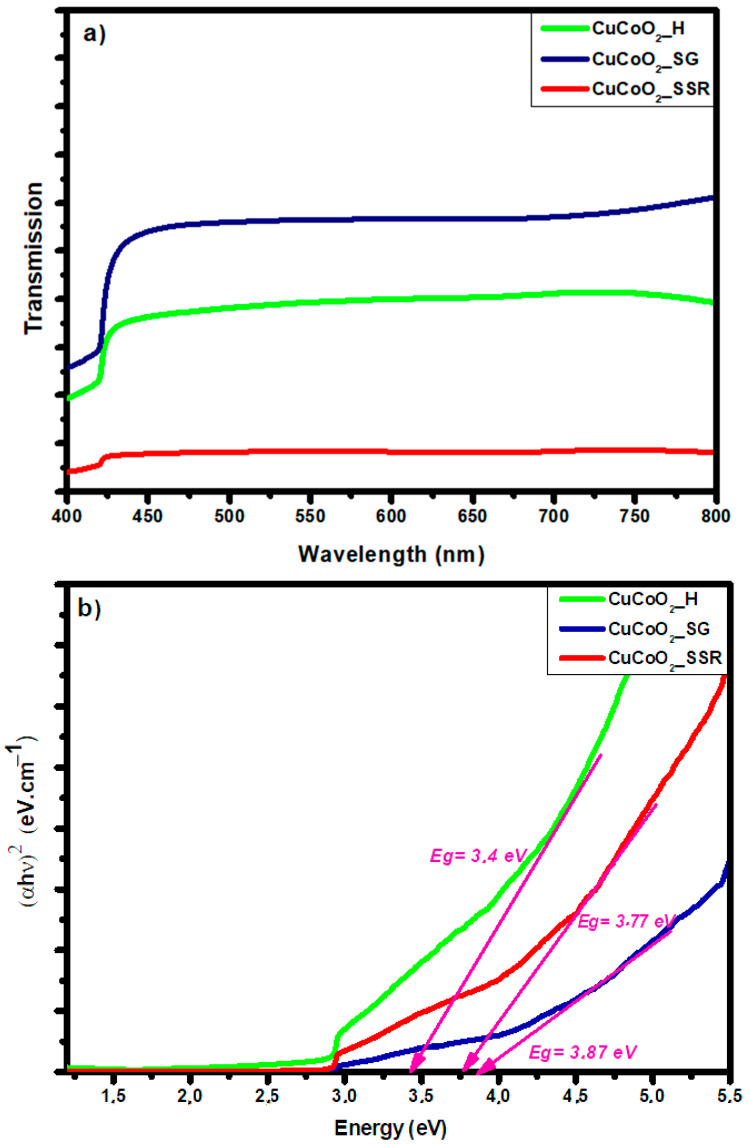
(**a**) Transmission and (**b**) Tauc’s plot for CuCoO_2__H, CuCoO_2__SG, and CuCoO_2__SSR precursor powders.

**Figure 11 nanomaterials-13-02312-f011:**
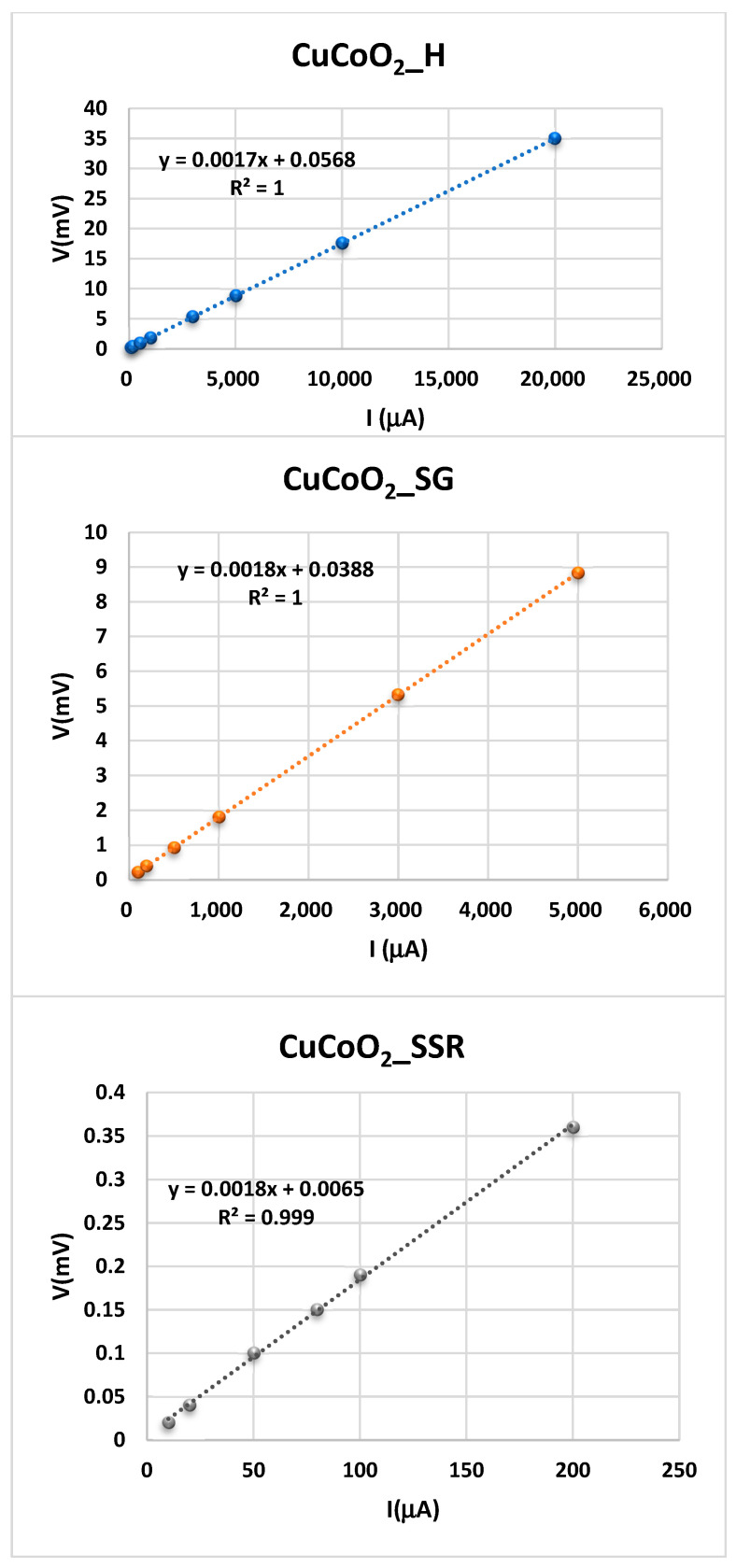
Variation of the potential between RE and S electrodes when a current intensity is given between WE and CE electrodes for the samples CuCoO_2_ thin films. (●), CuCoO_2__H, (●) CuCoO_2__SG, and (●) CuCoO_2__SSR, at ambient temperature.

**Figure 12 nanomaterials-13-02312-f012:**
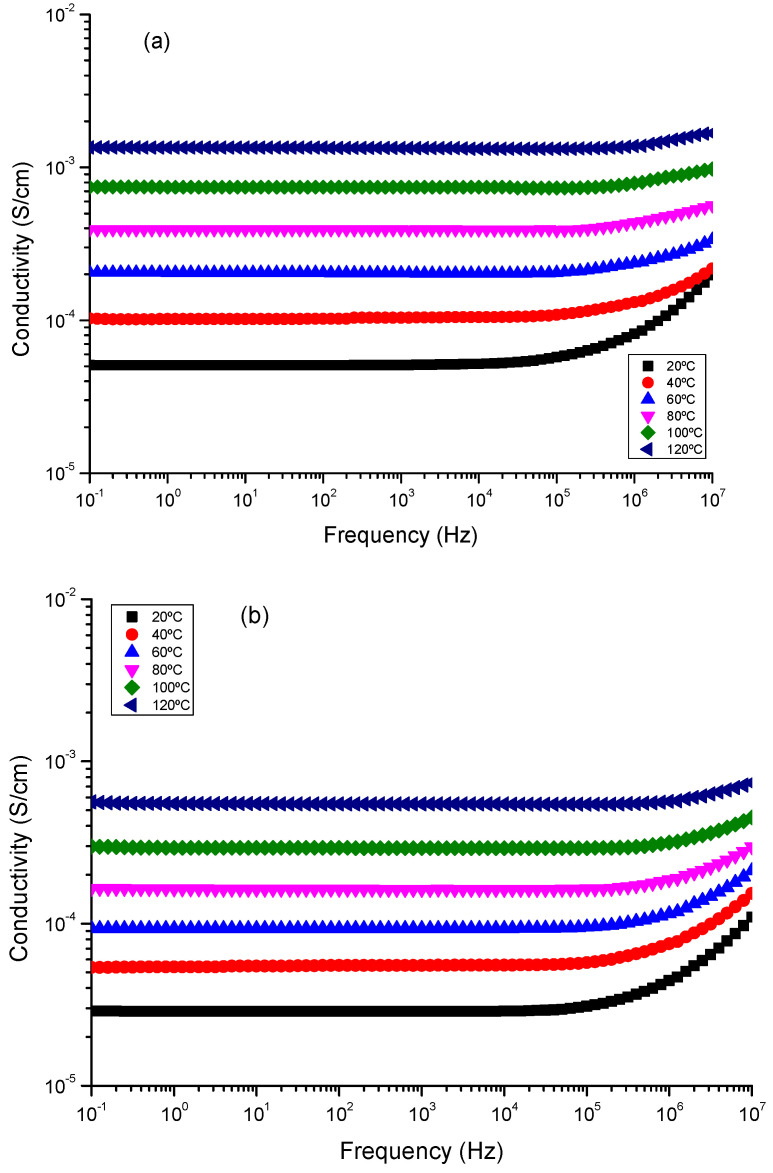
Bode diagrams of the conductivity for (**a**) CuCoO_2__SG, (**b**) CuCoO_2__SSR, and (**c**) CuCoO_2__H at different temperatures.

**Figure 13 nanomaterials-13-02312-f013:**
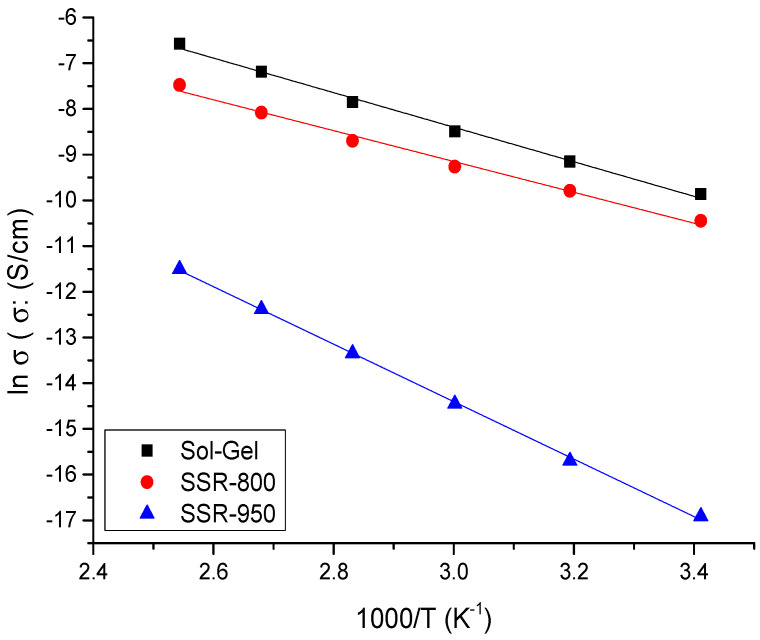
Temperature dependence of conductivity determined using Bode graphs for all samples examined.

**Figure 14 nanomaterials-13-02312-f014:**
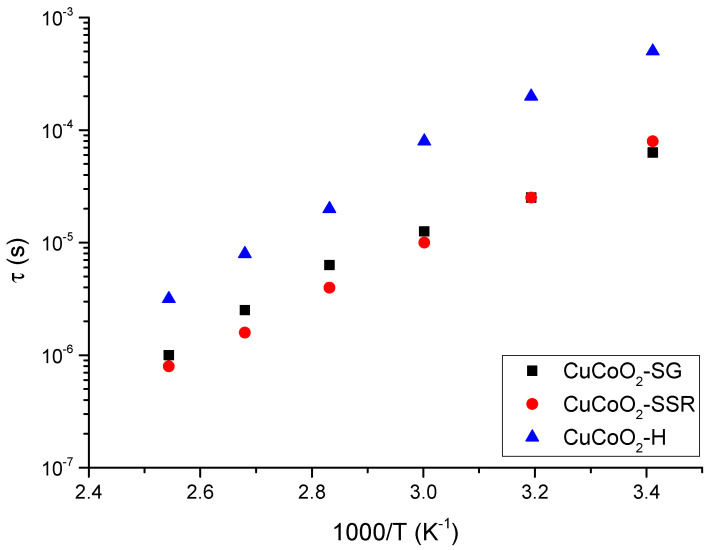
Variation of the sample relaxation time against the reciprocal of temperature for the delafossites investigated.

**Figure 15 nanomaterials-13-02312-f015:**
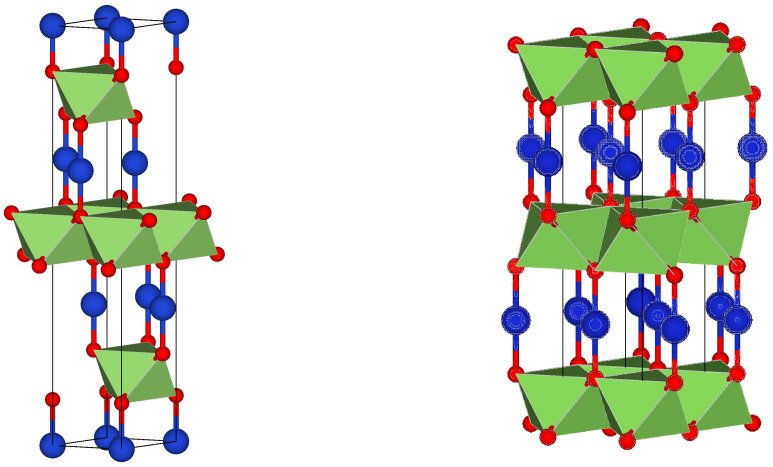
Trigonal and hexagonal structures of ${\mathrm{CuCoO}}_{2}$. The atoms of Cu are represented in blue; O atoms are presented in red; and the metal atoms lie inside the green octahedron.

**Figure 16 nanomaterials-13-02312-f016:**
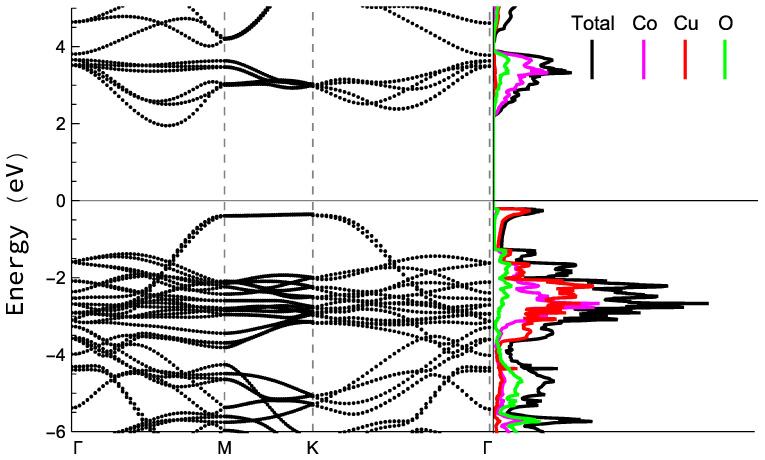
Bands structure (**left**) and states’ density (**right**) acquired by HSE simulations for the tetragonal structure. Energies are expressed with respect to the Fermi energy. Black line accounts for total DOS in the right part, and colored lines accounts for the contribution of every species to the density of states: Co atoms are represented by pink, Cu atoms by red, and O atoms by green.

**Table 1 nanomaterials-13-02312-t001:** The electrical properties of the CuCoO_2_ thin films determined at ambient temperature.

Samples	Thickness (nm)	Sheet Resistivity (ρ_s_) ×10^−3^ (Ω·cm)	Sheet Resistance (R_s_) (Ω)	Sheet Conductivity ×10^3^ (S·cm^−1^)
CuCoO_2__H	140 ± 5	0.114 ±0.002	8.16 ± 0.05	8.8 ± 0.2
CuCoO_2__SG	136 ± 5	0.105 ± 0.002	7.70 ± 0.04	9.5 ± 0.2
CuCoO_2__SSR	142 ± 5	0.116 ± 0.002	8.16 ± 0.04	8.6 ± 0.2

## Data Availability

The data presented in this study are available on request from the corresponding author.

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
