# Peer review of "The Structural and Electrochemical Properties of CuCoO2 Crystalline Nanopowders and Thin Films: Conductivity Experimental Analysis and Insights from Density Functional Theory Calculations"

_nanomaterials, 2023, doi:10.3390/nano13162312_

Round 1

Reviewer 1 Report

The Article is devoted to the synthesis and study of nanopowders and thin films of delafossite CuCoO2. The Authors synthesized delafossite powders by three different methods and prepared films on glass substrates. The properties of powders and films, in particular, the crystal structure, morphology and particle size distribution, optical and dielectric properties, have been studied by means of various methods. In addition, the results of a theoretical analysis of the structure and electronic states in the tetragonal and hexagonal phases of CuCoO2 are presented. The Article is of interest to the journal. The reviewer has the following comments on the text of the article:

1.                  The Experimental part only describes the synthesis of powders and films, and electrochemical impedance spectroscopy measurements. There is no description of other methods used. The description of the 4-probe method given in the Results and Duscussion section could also be moved to the Experimental part. 

2.                  In section 5. Optical Properties, the Authors present the results of the calculation of the band gap Eg using Tauc's plots. From Figure 9 it is seen that the intersection of the linear approximation with the abscissa axis will depend on the energy region for which the approximation is carried out. The errors associated with this are described in particular in: M.G. Brik, A.M. Srivastava, A.I. Popov. A few common misconceptions in the interpretation of experimental spectroscopic data. Optical Materials 127 (2022) 112276. https://doi.org/10.1016/j.optmat.2022.112276.

3.                  There are places in the Article that are not entirely clear. Some of them are marked in the attached pdf file.

 The Article needs to be revised.

 Minor editing of English language required

Author Response

Reviewer #1: The Article is of interest to the journal. The reviewer has the following comments on the text of the article:

  1. The Experimental part only describes the synthesis of powders and films, and electrochemical impedance spectroscopy measurements. There is no description of other methods used. The description of the 4-probe method given in the Results and Duscussion section could also be moved to the Experimental part. 

Response #: We took into account your recommendation and we moved the description of 4-probe method to the experimental part.

  1. 2.In section 5. Optical Properties, the Authors present the results of the calculation of the band gap Eg using Tauc's plots. From Figure 9 it is seen that the intersection of the linear approximation with the abscissa axis will depend on the energy region for which the approximation is carried out. The errors associated with this are described in particular in: M.G. Brik, A.M. Srivastava, A.I. Popov. A few common misconceptions in the interpretation of experimental spectroscopic data. Optical Materials 127 (2022) 112276. https://doi.org/10.1016/j.optmat.2022.112276.

Response #: After reading the recommended article, we found it to be an interesting investigation, and we were inspired by its content. As a result, we have cited it in our work. We would like to express our gratitude for bringing this paper to our attention. Thank you for your helpful suggestion.

Yours sincerely

The authors

Reviewer 2 Report

    This manuscript deals mainly with the synthesis of CuCoO2 crystalline nanopowders and thin films with hydrothermal, sol-gel and solid-state reactions as well as structural and electrochemical properties by means of XRD, SEM, HR-TEM, UV-visible and EIS methods. The prepared delafossite CuCoO2 show a well-ordered crystalline mixture comprising rhombohedral and hexagonal structures, exhibiting similar properties and favorable band gaps. The sample (CuCoO2_SG) prepared with sol-gel method exhibited superior conductivity to the other preparation methods and hence significant potential for solar cell applications, as supported by the DFT calculations. The conductivity values measured through plane increase with temperature, showing the order of σ(CuCoO2_SG) > σ(CuCoO2_SSR) > σ(CuCoO2_H).

 This manuscript performed interesting studies on the synthesis and structural and electrochemical characterizations of CuCoO2 crystalline nanopowders and thin films, and the results reflect the significant impact of the synthesis method and sample preparation on the optical and electrical properties of the materials, especially for solar cell applications. This manuscript was largely clearly written and well organized. So, this article can be recommended for publication in Nanomaterials. In addition, there are some revisions that could be considered by the authors.

1)      In Abstract, the relevant DFT calculations should also be mentioned.

2)      Please check and revise all the superscripts and subscripts in the whole text.

3)      In lines 42-43 of page 2, “since exhibiting” could be changed to “in view of”.

4)      In line 70 of page 3, the sentence seems unclear and needs to be modified.

5)      In Eq. (1) of page 5, the quantity “σ′ ” should be defined.

6)      In line 222 of page 13, “, that is caused by” may be changed to “due to”. In line 231 and also other positions, “such is” may be changed to “as”.

7)      In line 237 of page 14, “us” may be changed to “to”.

8)      In line 244 of page 16, “permit us observe” could be changed to “reveals”. In line 252 of this page, “depending on” could be changed to “with”.

9)      In line 268 of page 17, “was calculate” can be changed to “was calculated”.

10)In lines 311 and 312 of page 20, “showed” may be changed to “shown”, and “temperature increase and follow may be changed to with the increase of temperature.

11)  In line 329 of page 21, “we can say according the” can be changed to “according to the”. In line 339 of this page, “but is” could be changed to “but it is”. In lines 341 and 342, “That’s mean…if we want enhanced…” could be changed to “This means … for …”.

12)  In lines 343 and 344 of page 22, “on … destined for use in” could be changed to “to … in ”. In line 349 of this page, “properties of” can be changed to “properties of the materials”.

13)  In the first paragraph of page 23, comparison of the lattice constants from DFT calculations with the experimental data would be helpful. In line 377 of this page, “octahedral” may be changed to “octahedron”.

 14) In line 384 of page 23, the still underestimated bandgaps based on HSE06 functional should be briefly illustrated. In line 387 of this page, “which is why” could be changed to “so”.

The authors are adviced to check English of the whole text. 

Author Response

Reviewer #2:  This manuscript performed interesting studies on the synthesis and structural and electrochemical characterizations of CuCoO2 crystalline nanopowders and thin films, and the results reflect the significant impact of the synthesis method and sample preparation on the optical and electrical properties of the materials, especially for solar cell applications. This manuscript was largely clearly written and well organized. So, this article can be recommended for publication in Nanomaterials. In addition, there are some revisions that could be considered by the authors.

  1. In Abstract, the relevant DFT calculations should also be mentioned.

Response #: We appreciate your comment; the relevant DFT has been mentioned in the abstract.

  1. Please check and revise all the superscripts and subscripts in the whole text.

Response #: All the superscripts and subscripts has been checked in the whole revised manuscript.

  1. - In lines 42-43 of page 2, “since exhibiting” could be changed to “in view of”.

      - In line 70 of page 3, the sentence seems unclear and needs to be modified.

      - In Eq. (1) of page 5, the quantity “σ′ ” should be defined.

      - In line 222 of page 13, “, that is caused by” may be changed to “due to”. In line 231 and also other positions, “such is” may be changed to “as”.

      - In line 237 of page 14, “us” may be changed to “to”.

     - In line 244 of page 16, “permit us observe” could be changed to “reveals”. In line 252 of this page, “depending on” could be changed to “with”.

     - In line 268 of page 17, “was calculate” can be changed to “was calculated”.

     - In lines 311 and 312 of page 20, “showed” may be changed to “shown”, and “temperature increase and follow” may be changed to “with the increase of temperature”.

     - In line 329 of page 21, “we can say according the” can be changed to “according to the”. In line 339 of this page, “but is” could be changed to “but it is”. In lines 341 and 342, “That’s mean…if we want enhanced…” could be changed to “This means … for …”.

     - In lines 343 and 344 of page 22, “on … destined for use in” could be changed to “to … in ”. In line 349 of this page, “properties of” can be changed to “properties of the materials”.

    - In the first paragraph of page 23, comparison of the lattice constants from DFT calculations with the experimental data would be helpful. In line 377 of this page, “octahedral” may be changed to “octahedron”.

    - In line 384 of page 23, the still underestimated bandgaps based on HSE06 functional should be briefly illustrated. In line 387 of this page, “which is why” could be changed to “so”.

Response #: All the requested modifications have been made in the revised manuscript. Thank you for your attention. We have modified the list of references because we have include some of them reference and we have delete the reference   115 from the previous preprint and other duplicate references. For all this we have also rearranged the bibliography.

Sincerely yours

The authors

Reviewer 3 Report

1) what is the active phase with high-efficient electrochemical properties? 

2) HSE is quite accuracy to estimate the band gap, why the predicted value deviates so much from the experimental values?

Author Response

Reviewer 3

Comments and Suggestions for Authors

1) what is the active phase with high-efficient electrochemical properties? 

Response #: The change in dc conductivity of the samples at various temperatures may be determined from the plateau where the phase angle is zero o tends to zero. For frequencies where the phase is near zero we have a pure resistive impedance than can be attributed to the ionic conductivity alone.  This value is the active phase with high-efficiency electrochemical, at least respect the charge transport into the powders. These phenomena are observed for all the samples at frequencies below than 104 Hz.  Moreover, with rising temperature the frequency at which the point of equilibrium occurs moves to high frequencies, where a plateau in the Bode diagram more extensive from low to high frequencies can be observed, suggesting thermal activation of ionic transport. The constant value of conductivity suggests that the sample solely shows resistive contribution, and the quantity measured represents the sample's electrical conductivity.

2) HSE is quite accuracy to estimate the band gap, why the predicted value deviates so much from the experimental values?

Response #: HSE, hybrid functional can balance out in several cases the overestimation of Hartree-Fock and the underestimation of FDT method, especially in simple semiconductors. The studied delafossite is a complex system, an oxide wiith also d-electrons due to the transition metal, so the results may not be as good as in a simple semiconductor. In this case the underestimation is not very large.

Yours sincerely

The authors

Round 2

Reviewer 1 Report

Unfortunately, the Authors did not give an adequate response to the critical remark. The optical part of the Article remained unchanged. The Authors limited themselves to including a reference to the article cited in the remark in the list of references. It seems that the inclusion of this reference in the bibliography does not make sense. All that was required was to give some justification for the choice of the range of photon energies in which the linear approximation was carried out. How did the Authors choose the "linear part" of Tauc's plot?

The Article needs minor improvement.

 Minor editing of English language required

Author Response

Reviewer # Comments and Suggestions for Authors

Unfortunately, the Authors did not give an adequate response to the critical remark. The optical part of the Article remained unchanged. The Authors limited themselves to including a reference to the article cited in the remark in the list of references. It seems that the inclusion of this reference in the bibliography does not make sense. All that was required was to give some justification for the choice of the range of photon energies in which the linear approximation was carried out. How did the Authors choose the "linear part" of Tauc's plot?

Response #: Thank you very much for your comment. We apologize for the lack of details in the optical part about the procedure followed to determine the energy ban-gap from Tauc's plot. In the revised manuscript, we have provided additional information such as described in advance. The authors think that in the new version a valid argument has been given to choose the appropriate energy band.

The linear part is selected in a region where the absorption coefficient is relatively constant and not influenced by other factors like impurities or defects.

The selection process involves visually inspecting the Tauc's plot and identifying a portion of the curve that appears to be linear. This is usually done by locating a range of higher photon energies where the absorption coefficient starts to increase linearly with energy.

In Tauc's plot, the "linear part" is selected by examining the absorption data at higher photon energies. Typically, this linear region occurs at higher energy levels, where the absorption is predominantly governed by indirect transitions. The linear portion represents the onset of the absorption edge and is used to determine the bandgap energy of a semiconductor material. By extrapolating the linear part to the x-axis, the bandgap energy can be estimated from the intercept.

In the new manuscript we have added the following paragraph:

In the Tauc's plot, the "linear part" is selected by examining the absorption data at higher photon energies, where the absorption is predominantly governed by indirect transitions being its absorption coefficient practically constant. In our study the calculation of band gap values has an error of ±0,10 eV was obtained taking the linear part of the curve (between 4 and 4,5 eV), fitting these points to a straight line and extrapolating this line until it intersects the base line (OX axis). The intersection value (in eV) is the direct band gap according to Tauc's model78. In our study the values obtained for our delafossite material are 3,51±0,10 eV, 3,77±0,12 eV and 3,87±0,10 eV for CuCoO2-H, CuCoO2-SSR and CuCoO2-SG, respectively. Depending on the number of points chosen in the range considered it works as if it were a transparent layer with a certain uncertainty. This suggests that delafossite is likely to be a good transmitter of charge carriers, leading to a higher band gap value around of 3.5 eV.

Sincerely yours

The authors